# [^68^Ga]Ga-DOTAGA-Glu(FAPi)_2_ Shows Enhanced Tumor Uptake and Theranostic Potential in Preclinical PET Imaging

**DOI:** 10.3390/diagnostics14182024

**Published:** 2024-09-13

**Authors:** Julie van Krimpen Mortensen, Simona Mattiussi, Lars Hvass, Emilie Graae Lund, Vladimir Shalgunov, Frank Roesch, Umberto Maria Battisti, Matthias Manfred Herth, Andreas Kjaer

**Affiliations:** 1Department of Clinical Physiology and Nuclear Medicine, Copenhagen University Hospital—Rigshospitalet, 2100 Copenhagen, Denmark; julie.mortensen@sund.ku.dk (J.v.K.M.); lars.hvass@sund.ku.dk (L.H.); emilie.lund@sund.ku.dk (E.G.L.); vladimir.shalgunov@sund.ku.dk (V.S.); matthias.herth@sund.ku.dk (M.M.H.); 2Cluster for Molecular Imaging, Copenhagen University Hospital—Rigshospitalet & Department of Biomedical Sciences, University of Copenhagen, 2100 Copenhagen, Denmark; 3Department of Drug Design and Pharmacology, University of Copenhagen, Jagtvej 160, 2100 Copenhagen, Denmark; simona.mattiussi@sund.ku.dk (S.M.); umberto.battisti@sund.ku.dk (U.M.B.); 4Department of Chemistry-TRIGA Site, Johannes Gutenberg University Mainz, 55128 Mainz, Germany; germanyfroesch@uni-mainz.de

**Keywords:** [^68^Ga]Ga-DOTAGA-Glu(FAPi)_2_, fibroblast activation protein inhibitors, preclinical PET imaging, gallium-68, theranostics

## Abstract

The use of fibroblast activation protein inhibitors (FAPis) for positron emission tomography (PET) imaging in cancer has garnered significant interest in recent years, yielding promising results in preclinical and clinical settings. FAP is predominantly expressed in pathological conditions such as fibrosis and cancer, making it a compelling target. An optimized approach involves using FAPi homodimers as PET tracers, which enhance tumor uptake and retention, making them more effective candidates for therapy. Here, a UAMC-1110 inhibitor-based homodimer, DOTAGA-Glu(FAPi)_2_, was synthesized and radiolabeled with gallium-68, and its efficacy was evaluated in vivo for PET imaging in an endogenously FAP-expressing xenografted mouse model, U87MG. Notably, 45 min post-injection, the mean uptake of [^68^Ga]Ga-DOTAGA-Glu(FAPi)_2_ was 4.7 ± 0.5% ID/g in the tumor with low off-target accumulation. The ex vivo analysis of the FAP expression in the tumors confirmed the in vivo results. These findings highlight and confirm the tracer’s potential for diagnostic imaging of cancer and as a theranostic companion.

## 1. Introduction

Despite significant treatment advancement, cancer continues to pose a major global challenge, accounting for nearly 10 million deaths worldwide in 2022. In that year alone, there were an estimated 20 million new cancer cases, emphasizing the increasing disease burden [1]. Early detection and appropriate treatment are crucial for reducing the cancer burden, which can be assisted by advancing diagnostic and therapeutic strategies [2].

Fibroblast activation protein (FAP) is a type II transmembrane serine protease that has emerged as a promising target in various cancer types due to its overexpression in several malignancies, specifically epithelial cancers [3]. FAP cleaves the peptide bond between the proteinogenic amino acid proline and other amino acids, thereby altering various bioactive molecules. Among other proteases with similar activity are dipeptidyl peptidase (DDP) 4, 6, 8, and 9, where FAP is closely related to DPP4 [4]. FAP is predominantly expressed in activated fibroblasts such as cancer-associated fibroblasts (CAFs) present in the tumor stroma, where they actively remodel the extracellular matrix (ECM) and influence tumor behavior. Thus, targeting FAP addresses the tumor microenvironment (TME), compared to other tumor targets that generally target the tumor cells directly [5]. This potentially renders the utility of FAP as a much broader diagnostic and therapeutic target as it can be applied to many cancer types [6,7]. Recent studies have highlighted the role of FAP in the TME, where it not only contributes to ECM remodeling but also influences tumor growth, invasion, and immune evasion [8,9,10,11]. The overexpression of FAP in CAFs is associated with poor prognosis in several cancers, making it a critical target for therapeutic intervention [9]. Moreover, FAP’s enzymatic activities extend beyond its role in cancer, as it is also involved in various pathological conditions such as fibrosis and arthritis [12]. Furthermore, FAP expression in normal adult tissues, where resting fibroblast is present, is minimal, underlying the suitability of FAP as a target for both diagnostic and therapeutic purposes [4,5].

The identification of targeting agents for FAP is of increasing interest driven by the idea of selectively targeting FAP, while DPP4 remains unaffected [4]. The specific targeting of FAP could contribute to targeting the tumor and the TME. FAP inhibitors (FAPis) have been exploited as small-molecule radioligands designed to specifically target FAP with high binding affinity and are actively being investigated for both positron emission tomography (PET) imaging and potential radioligand therapies across various cancer types [5,13].

FAPi dimers represent an innovative approach to enhance tumor uptake, retention, and imaging contrast compared to their monomeric counterparts, optimizing their potential for both imaging and radionuclide therapy [14]. The philosophy of dimerization in PET tracers is grounded in enhancing biological interactions using dual binding sites. This approach has shown considerable promise in improving tumor uptake, retention, and predictive value for therapeutic efficacy [15,16]. Their extended tumor retention makes dimers particularly advantageous for theranostic purposes, especially when paired with radioisotopes like 177-Lutetium (^177^Lu), which have a prolonged half-life. This combination enhances therapeutic efficacy by leveraging the prolonged retention characteristic of the molecule [15]. Combining diagnostic imaging with therapeutic intervention within a single molecular agent holds immense clinical value, offering personalized treatment strategies and the real-time monitoring of therapeutic response [17].

Recently, Martin and colleagues synthesized and evaluated DOTAGA-Glu(FAPi)_2_ (Figure 1), a promising UAMC-1110 inhibitor-based homodimer [18].

DOTAGA-Glu(FAPi)_2_ exhibits a very high affinity for FAP (IC50(FAP)/nM 0.26 ± 0.04) [18]. A high affinity for prolyl endopeptidase (PREP) or DPP, ubiquitously expressed in healthy tissue, would reduce tumor selectivity and result in a lower tumor-to-background ratio. Therefore, DOTAGA-Glu(FAPi)_2_ needs to demonstrate high selectivity for FAP over PREP and DPP. It has been found to have higher selectivity than DOTAGA.(SA.FAPi)_2_, another promising FAPi derivative [19]. This indicates that it may also have superior properties in vivo compared to other dimeric FAP tracers, making it suitable for further investigations. In this study, we evaluated the potential of gallium-68 (^68^Ga)-labeled DOTAGA-Glu(FAPi)_2_ ([^68^Ga]Ga-DOTAGA-Glu(FAPi)_2_) for the in vivo PET imaging of mice xenografted with an endogenously FAP-expressing human glioblastoma cell line, aiming to demonstrate enhanced tumor uptake and retention, thereby supporting its use as an effective theranostic agent. By leveraging the unique properties of FAPi dimers, such as their enhanced tumor retention [20], we aimed to explore the potential of the ^68^Ga-labeled molecule as a theranostic companion.

## 2. Materials and Methods

### 2.1. Organic Synthesis

DOTAGA-Glu(FAPi)_2_ was synthesized according to published procedures [18]. The synthesis process is visualized in Figure 2.

#### 2.1.1. Synthesis of Glu.(FAPi)_2_

In our synthesis, we prepared tert-Butyl ((S)-1,5-bis((4-((4-((2-((S)-2-cyano-4,4-difluoropyrrolidin-1-yl)-2-oxoethyl)caRbamoyl)-quinolin-6-yl)oxy)butyl)amino)-1,5-dioxopentan-2-yl)carbamate (**3**,Boc-Glu.(FAPi)_2_).

N-tert-Butoxycarbonyl-L-glutamic acid **2** (Boc-Glu-OH, 118 mg, 0.476 mmol, 1.00 eq), 1-hydroxybenzotriazole (HOBt, 167 mg, 1.233 mmol, 2.59 eq), and 1-ethyl-3-(3-dimethylaminopropyl) carbodiimide hydrochloride (EDC*HCl, 238 mg, 1.243 mmol, 2.61 eq) were dissolved in dry N,N-dimethylformamide (DMF, 3 mL). N,N-diisopropylethylamine (DIPEA, 41 μL, 0.238 mmol, 0.50 eq) was then added under argon condition. The solution was stirred at room temperature for one hour, during which it turned yellow. Subsequently, a solution of FAPi-NH_2_ **1** (446 mg, 0.952 mmol, 2.00 eq) and DIPEA (412 μL, 2.380 mmol, 5.00 eq) in DMF (3 mL) was added. The reaction mixture was stirred at room temperature overnight, and the solvent was removed in vacuo. The solution was then diluted with water (5 times the volume of the organic solvent), and the aqueous phase was extracted with EtOAc (3 × 30 mL). The organic phase was dried over Na_2_SO_4_, and the solvent was removed under reduced pressure. After column chromatography (CH_2_Cl_2_/MeOH, 95:0.5–10) r*f*: 0.375 **3** was obtained as a colorless oil (350 mg, 0.326 mmol, 69%). ^1^H-NMR (400 MHz, MeOD-d4): δ [ppm] 8.71 (t, *J* = 4.6 Hz, 2H), 7.93 (d, *J* = 9.2 Hz, 2H), 7.89 (d, *J* = 2.7 Hz, 2H), 7.53 (dd, *J* = 4.5, 2.8 Hz, 2H), 7.42 (dd, *J* = 9.2, 2.7 Hz, 2H), 5.15 (dt, *J* = 9.5, 2.8 Hz, 2H), 4.40–3.87 (m, 13H), 3.30–3.20 (m, 3H), 3.02–2.74 (m, 4H), 2.24 (t, *J* = 7.5 Hz, 2H), 1.87 (dq, *J* = 12.5, 6.6 Hz, 5H), 1.72 (q, *J* = 7.0 Hz, 4H), 1.38 (s, 9H). MS (ESI^+^): *m*/*z* (%) = 537.8 (95, [M+H]^2+^), 1074.4 (72, [M+H]^+^), 1075.4 (50, [M+H]^+^), 1076.4 (20, [M+H]^+^) calculated for C_52_H_59_F_4_N_11_O_10_: 1073.44 [M].

(S)-2-Amino-N1,N5-bis(4-((4-((2-((S)-2-cyano-4,4-difluoropyrrolidin-1-yl)-2-oxoethyl)carbamoyl)-quinolin-6-yl)oxy)butyl)pentanediamide (**4**, Glu.(FAPi)_2_) was then prepared from Boc-Glu.(FAPi)_2_ **3** (350 mg, 0.326 mmol, 1.00 eq), which was dissolved in dry 1,4-dioxane (5 mL) under an argon atmosphere. At 0 °C, 4 M hydrochloric acid (HCl) in 1,4-dioxane (1 mL, 4.24 mmol, 13.00 eq) was added, resulting in the formation of a white precipitate. The reaction was allowed to warm to room temperature, and after 3 h, the solvent was completely removed in vacuo, yielding Glu.(FAPi)_2_ as a yellowish solid **4** (311 mg, 0.308 mmol, 95%). ^1^H NMR (600 MHz, MeOD) δ 9.02 (dd, *J* = 8.3, 5.4 Hz, 2H), 8.21 (dd, *J* = 8.4, 2.7 Hz, 2H), 8.14 (dd, *J* = 9.3, 2.7 Hz, 2H), 7.99 (dt, *J* = 10.5, 4.1 Hz, 2H), 7.77 (ddd, *J* = 8.9, 5.8, 2.6 Hz, 2H), 5.16 (td, *J* = 10.0, 3.1 Hz, 2H), 4.48–4.03 (m, 12H), 3.92 (t, *J* = 6.3 Hz, 1H), 3.07–2.72 (m, 6H), 2.43 (p, *J* = 7.4 Hz, 2H), 2.11 (hept, *J* = 7.5 Hz, 2H), 1.95 (dq, *J* = 21.1, 6.9 Hz, 4H), 1.78 (dp, *J* = 30.0, 7.3 Hz, 4H). MS (ESI^+^): *m*/*z* (%) = 325.6 (100, [M-Boc^+^H]^3+^), 487.8 (85, [M+H]^2+^), 974.3 (45, [M+H]^+^), calculated for C_47_H_51_F_4_N_11_O_8_: 973.39 [M].

#### 2.1.2. Synthesis of DOTAGA.Glu.(FAPi)_2_

We prepared tri-tert-Butyl 2,2′,2″-(10-(5-(((S)-1,5-bis((4-((4-((2-((S)-2-cyano-4,4-difluoropyrrolidin-1-yl)-2-oxo-ethyl)carbamoyl)quinolin-6-yl)oxy)butyl)amino)-1,5-dioxopentan-2-yl)amino)-1-(tert-butoxy)-1,5-dioxopentan-2-yl)-1,4,7,10-tetraazacyclododecane-1,4,7-triyl)triacetate (**6**, DOTAGA(^t^Bu)_4_.Glu.(FAPi)_2_).

DOTAGA(^t^Bu)_4_ **5** (40 mg, 0.057 mmol, 1.15 eq), 1-[bis(dimethylamino)methylene]-1H-1,2,3-triazolo [4,5-b]pyridinium 3-oxide hexafluorophosphate (HATU, 21.6 mg, 0.057 mmol, 1.15 eq) and DIPEA (10.7 μL, 0.062 mmol, 1.25 eq) were dissolved in dry DMF (1 mL) under argon atmosphere and stirred at room temperature for one hour. The solution changed color from clear pink to yellow. Then, the solution of Glu.(FAPi)_2_ (50 mg, 0.05 mmol, 1.00 eq) and DIPEA (21.4 μL, 0.124 mmol, 2.50 eq) in dry DMF (2 mL) was added. The reaction mixture reacted for 2 h, and the solvent was removed in vacuo. The crude product was purified by reverse phase (gradient from 0 to 100% MeCN, t*_R_* = 12.4 min) and a yellow solid was obtained. MS (ESI^+^): *m*/*z* (%) = 415.23 (75, [M+H]^4+^), 553.46 (100, [M+H]^3+^), 829.28 (80, [M+H]^2+^), 830.17 (20, [M+H]^2+^), 1656.85 (87, [M+H]^+^), 1657.85 (85, [M+H]^+^), 1658.85 (43, [M+H]^+^), 1659.86 (15, [M+H]^+^), calculated for C_82_H_113_F_4_N_15_O_17_: 1655.84 [M].

2,2′,2″-(10-(4-(((S)-1,5-bis((4-((4-((2-((S)-2-Cyano-4,4-difluoropyrrolidin-1-yl)-2-oxoethyl)carbamoyl)quinolin-6-yl)oxy)butyl)amino)-1,5-dioxopentan-2-yl)amino)-1-carboxy-4-oxobutyl)-1,4,7,10-tetraazacyclododecane-1,4,7-triyl)triacetic acid (**7**, DOTAGA.Glu.(FAPi)_2_) was also obtained.

DOTAGA(^t^Bu)_4_.Glu.(FAPi)_2_ **6** was dissolved in TFA (601 μL), MeCN (70 μL), triisopropylsilane (TIPS, 36 μL) and H_2_0 (18 μL) and stirred at room temperature for 4 h.

The crude product was purified by semipreparative RP-HPLC (35% MeCN in 30 min, tR = 31 min), and **7** was obtained as a yellow solid (20.0 mg, 0.014 mmol, 28%). MS (ESI^+^): *m*/*z* (%) = 359.1 (55, [M+H]^4+^), 478.4 (100, [M+H]^3+^), 716.9 (40, [M+H]^2+^), 1432.40 (20, [M+H]^+^), calculated for C_66_H_81_F_4_N_15_O_17_: 1431.59 [M].

### 2.2. Labeling

Gallium-68 chloride was obtained by eluting the ^68^Ge/^68^Ga generator (Galliapharm, Eckert & Ziegler, Berlin, Germany) with 0.1 M HCl. Ammonium acetate buffer (0.1 M, pH 5.5), and a DOTAGA-Glu(FAPi)_2_ stock solution (2 mg/mL) for labeling was prepared using water for ultratrace analysis (Merck, Darmstadt, Germany). [^68^Ga]GaCl_3_ solution in 0.1 M HCl (500 µL, ca. 200 MBq ^68^Ga) was mixed with ammonium acetate buffer (80 µL) and DOTAGA-Glu-(FAPi)_2_ stock solution (4 µL). The resulting mixture was placed on an agitating mixer and shaken at 600 rpm for 5 min at 60 °C. Afterward, the reaction mixture was diluted with 5 mL water and passed through a Strata-X 33 µm polymeric reversed-phase cartridge (Phenomenex, Torrance, CA, USA). The cartridge was rinsed with more water (0.3 mL) and eluted with absolute ethanol (0.5 mL). Ethanol was evaporated under nitrogen flow and gentle heating to 40 °C. The residue was resolubilized in an aliquot of fresh ethanol (20 µL) and further diluted with phosphate buffer (0.1 M, pH 7) to a total volume of 2 mL.

Radiochemical conversion (RCC) for the ^68^Ga-labeling process and radiochemical purity (RCP) of the formulated [^68^Ga]Ga-DOTAGA-Glu(FAPi)_2_ were assessed by radio-HPLC.

### 2.3. In Vivo Evaluation

Six-week-old female NMRI nude mice were purchased (Janvier, Le Genest-Saint-Isle, France) and housed in groups of 4–8 mice in individually ventilated cages under regular lighting conditions. They were fed a standard pathogen-free pellet diet and provided water ad libitum. All animal experiments were approved by The Animal Experiments Inspectorate in Denmark (2021-15-0201-01041). The animals were acclimatized for two weeks before study initiation.

To establish a tumor model, the mice were subcutaneously inoculated on the right flank with a human xenograft glioblastoma U87MG (HTB-14, ATCC, Manassas, VA, USA). These cells were cultivated in Dulbecco’s modified Eagle’s medium (DMEM) supplemented with fetal bovine serum (FBS) and penicillin–streptomycin (Pen/Strep). Tumor growth was monitored two times weekly using calipers.

After adequate tumor establishment, the animals were injected intravenously in the tail vein with 4.49 ± 0.04 MBq [^68^Ga]Ga-DOTAGA-Glu(FAPi)_2_ in a 100 μL phosphate buffer (0.1 M, pH 7) with 1% ethanol (*v*/*v*). PET/CT imaging (Inveon, Siemens Healthineers, Knoxville, TN, USA) was performed immediately after injection, and PET data were acquired for 3000 s. The imaging protocol included a dynamic framing setup, with five consecutive frames of 600 s each, resulting in average time intervals of 0–10, 10–20, 20–30, 30–40, and 40–50 min post-injection. During imaging, body temperature was maintained using a heated platform.

PET images were reconstructed using the 3D-OSEM/SP-MAP algorithm, and attenuation correction was performed using the co-registered CT images. The activity concentration was decay-corrected to the time of injection and quantified using Inveon Research Workplace (IRW). Regions of interest were manually drawn on CT images to delineate organs and derive volume and activity [21]. Subsequently, the percentage of injected dose per gram (% ID/g) was calculated for tumor, blood, liver, kidney, and muscle tissues for each frame.

All graphical representations of the data were generated using Prism version 10.1.0 (1994–2023, GraphPad Software, LLC, Boston, MA, USA). The results were expressed as the mean ± standard error of the mean (SEM).

### 2.4. Ex Vivo Evaluation

All tumor xenografts were resected and fixed in 4% paraformaldehyde. Following fixation, the tissues were embedded in paraffin and sectioned into slices, which were then deparaffinized and rehydrated to prepare them for immunohistochemical (IHC) analysis. The tissue sections were incubated for 24 h at 4 °C with the recombinant anti-FAP ɑ antibody (ab218164 Abcam, Cambridge, UK) at a dilution of 1:50. The secondary antibody was then applied, followed by staining with DAB+ chromogen, which produced a brown precipitate, indicating positive immunoreactivity. The sections were counterstained with hematoxylin to visualize cell nuclei and then dehydrated and mounted for microscopy. The specific binding of the antibody was ensured through the inclusion of negative controls.

High-quality virtual slides of the IHC results were obtained using the ZEISS Axio Scan Z1 whole slide scanner (ZEISS, Oberkochen, Germany), which allowed for the capture of detailed, high-resolution images of the entire tissue sections. Image analysis and quantification were performed using ZEISS ZEN 3.8 software, enabling the assessment of the FAP expression.

## 3. Results

### 3.1. Organic Synthesis

DOTAGA-Glu-(FAPi)_2_ was obtained with a purity of 99% as initially reported by Martin et al. [18]. The identity of the compound was confirmed by LC-MS, and the purity was defined by HPLC-MS (Appendix A).

### 3.2. Radiosynthesis

[^68^Ga]Ga-DOTAGA-Glu-(FAPi)_2_ was successfully obtained in 61% isolated radiochemical yield (RCY) [22]. Labeling and formulation took about 20 min. The RCC of ^68^Ga chelation was 96%, and the RCP of the final formulated tracer was 88% (Appendix A). RCP was stable for at least 1 h post-formulation.

### 3.3. In Vivo Evaluation

Tumor-bearing mice (*n* = 4) were successfully imaged with [^68^Ga]Ga-DOTAGA-Glu-(FAPi)_2_ PET/CT. A representative dynamic PET/CT image series of the in vivo cancer model is presented in Figure 3 showing the radioactive uptake at 5, 15, 25, 35, and 45 min post-injection.

The mean uptake at 45 min post-injection was 4.7 ± 0.5% ID/g in the tumor, 3.5 ± 0.2% ID/g in blood, 1.2 ± 0.1% ID/g in muscle, 2.2 ± 0.2% ID/g in kidneys, and 2.7 ± 0.1% ID/g in the liver and was cleared mainly by the kidneys (Figure 4a). Detailed uptake data at the various time points post-injection are presented in Appendix A.

The mean tumor-to-background ratios at 45 min post-injection were 1.4 ± 0.3 for the tumor-to-blood ratio (TBR), 3.9 ± 0.4 for the tumor-to-muscle ratio (TMR), 2.6 ± 0.1 for the tumor-to-kidney ratio (TKR), and 1.7 ± 0.2 for the tumor-to-liver ratio (TLR). The tumor-to-organ ratios at different time intervals can be seen in Figure 4b. The detailed ratios over the imaging period are provided in Appendix A.

### 3.4. Ex Vivo Evaluation

The ex vivo IHC analysis of the resected tumor tissue revealed distinctive FAP expression. The observed immunopositivity in the tumor cells was consistent with the expected endogenous expression of FAP. The brown DAB staining, which marks FAP-positive areas, was visible across the tumor cells, with additional staining detected in the stromal regions. Figure 5 illustrates these findings, with FAP immunopositivity depicted in brown, while the cell nuclei are counterstained in blue with hematoxylin.

## 4. Discussion

Here, we demonstrate the suitability of [^68^Ga]Ga-DOTAGA-Glu(FAPi)_2_ for in vivo PET imaging. Previous studies have evaluated the therapeutic potential of the molecule labeled with ^177^Lu [18,20,23]. To the authors’ knowledge, this is the first in vivo evaluation of [^68^Ga]Ga-DOTAGA-Glu(FAPi)_2_.

The tumor uptake of [^68^Ga]Ga-DOTAGA-Glu-(FAPi)_2_ showed increased accumulation with high tumor retention at 45 min post-injection, whereas the blood, kidney, and liver uptake decreased over the same period, depicting rapid clearance as the compound cleared renally. The kinetic profile of [^68^Ga]Ga-DOTAGA-Glu-(FAPi)_2_ is explained considering its low lipophilicity, estimated by measuring the logD_7.4_ value using the shake flask method with *n*-octanol and phosphate-buffered saline (pH 7.4). The logD_7.4_ of −2.5 ± 0.05 indicates good hydrophilicity, leading to renal excretion over biliary excretion [9]. The hydrophilicity is comparable to the monomer [^68^Ga]Ga-DOTA.SA.FAPi (−2.7 ± 0.1) ([24]) but with the advantage of high tumor uptake due to its dimeric nature. Moreover, the compound is more hydrophilic than the first-generation dimer [^68^Ga]Ga-DOTAGA.(SA.FAPi)_2_ (−2.0 ± 0.1) ([25])**_,_** resulting in lower background uptake and higher tumor-to-background image contrast.

During the initial 15 min post-injection, a small accumulation of [^68^Ga]Ga-DOTAGA-Glu(FAPi)_2_ was observed in the muscle tissue, most likely due to blood perfusion; however, uptake remained low.

An elevated tumor-to-muscle ratio was consistently observed throughout the imaging period, indicating specific tumor tissue accumulation. Additionally, the continuous increase in tumor-to-blood, tumor-to-kidney, and tumor-to-liver ratios further validates the compound’s sensitivity. These findings emphasize the efficacy of the tracer for accurate diagnostics and imaging. A comparison to monomeric counterparts suggests that dimerization enhances contrast and uptake, potentially improving diagnostic precision and therapeutic outcomes [14].

The ex vivo analysis of the FAP expression in the tumors corroborates the in vivo findings, confirming the expected endogenous expression of FAP within these tumor cells. The distinct FAP immunopositivity observed in both the tumor cells and the stromal component underscores the effectiveness of [^68^Ga]Ga-DOTAGA-Glu(FAPi)_2_ in specifically targeting FAP-positive tumors. This specificity is crucial for accurate imaging and subsequent therapeutic targeting, as it ensures that the tracer is selectively accumulating in FAP-expressing tumor cells rather than in non-target tissues. Importantly, the selective uptake of [^68^Ga]Ga-DOTAGA-Glu(FAPi)_2_ in FAP-expressing tissues, rather than non-target tissues, highlights its precision in targeting and imaging FAP-expressing tumors, making it a promising tool for the management of FAP-positive cancers.

Radiolabeling with ^68^Ga is advantageous due to its physical properties, including a short half-life of 68 min, which matches the kinetics and rapid tumor accumulation of [^68^Ga]Ga-DOTAGA-Glu(FAPi)_2_. Exploiting the compatibility of ^68^Ga decay kinetics with the circulation and tumor uptake kinetics of the FAPi molecule resulted in excellent image quality within 45 min. Furthermore, the relatively short half-life of ^68^Ga facilitates timely imaging procedures while minimizing radiation exposure to patients, thus making it an ideal choice for routine clinical PET imaging [26,27]. For this reason, [^68^Ga]Ga-DOTAGA-Glu(FAPi)_2_ could represent a better diagnostic compared to [^68^Ga]Ga-FAPI-04, a commonly used tracer that targets FAP. The latter has demonstrated a strong uptake in various types of cancer with improved biodistribution, tumor-to-background contrast, and faster kinetics compared to [^18^F]FDG. However, its relatively short tumor retention time makes it unsuitable for theranostic applications [28].

Long tumor retention is advantageous for therapeutic purposes, for instance, when labeling with long-lived radioisotopes like ^177^Lu [29]. Yadav et al. [23] demonstrated the efficacy and safety of [^177^Lu]Lu-DOTAGA-FAPi dimer treatment, yielding promising outcomes. This study compared the dimer and monomer forms, revealing significantly higher absorbed doses within the tumor with the dimer formulation.

The development of FAPi dimers represents a significant advancement in molecular imaging and targeted cancer therapy [30]. By leveraging the increased tumor uptake, retention, and imaging contrast achieved through dimerization, these agents not only improve diagnostic accuracy but also present a viable approach for developing radiotherapeutic analogs of FAPi-based molecules [31].

## 5. Conclusions

We successfully demonstrated the applicability of [^68^Ga]Ga-DOTAGA-Glu(FAPi)_2_ PET in vivo, highlighting its potential as a diagnostic tracer. Compared to existing tracers such as [^68^Ga]Ga-FAPI-04, [^68^Ga]Ga-DOTAGA-Glu(FAPi)_2_ offers the advantage of longer tumor retention and improved tumor-to-background contrast, making it a promising candidate for both diagnostic and therapeutic applications.

Given the dimers’ extended tumor retention time, this tracer also has significant potential as a theranostic agent when labeled with, e.g., ^177^Lu. The dimeric form of FAPi may provide a better predictive value for therapeutic efficacy than monomeric forms, underscoring its promise achieved through dimerization; thus, [^68^Ga]Ga-DOTAGA-Glu(FAPi)_2_ shows potential as an effective theranostic companion.

## Figures and Tables

**Figure 1 diagnostics-14-02024-f001:**
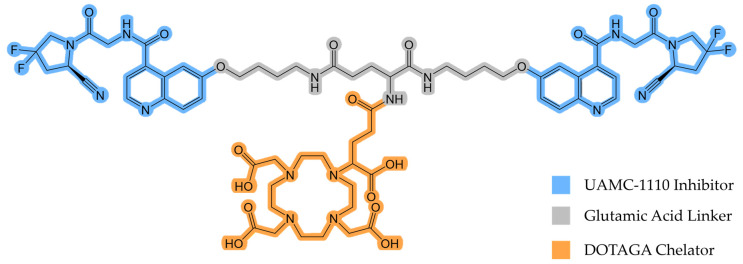
The chemical structure of DOTAGA-Glu(FAPi)_2_.

**Figure 2 diagnostics-14-02024-f002:**
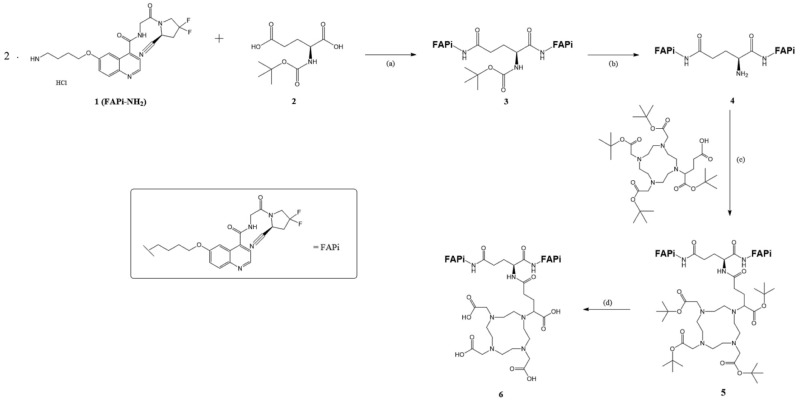
Organic synthesis of DOTA.GA.Glu.(FAPi)_2_ **6**: (**a**) HOBt, EDC*HCl, DIPEA, DMF, RT, overnight, 69%; (**b**) 4 M HCl in 1,4-dioxane, 1,4-dioxane, 0 °C-RT, 3 h, 95%; (**c**) HATU, DIPEA, DMF, and RT, overnight, 98%; (**d**) TFA:MeCN:TIPS:H_2_O (85:10:5:2.5), RT, 5 h, 28%.

**Figure 3 diagnostics-14-02024-f003:**
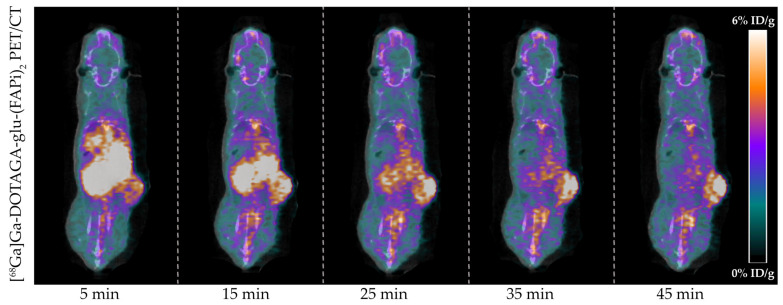
In vivo [^68^Ga]Ga-DOTAGA-Glu-(FAPi)_2_ PET/CT imaging of U87MG cancer model. Representative images of the radioactive uptake are visualized from 0 to 6%ID/g at 5, 15, 25, 35, and 45 min post-injection.

**Figure 4 diagnostics-14-02024-f004:**
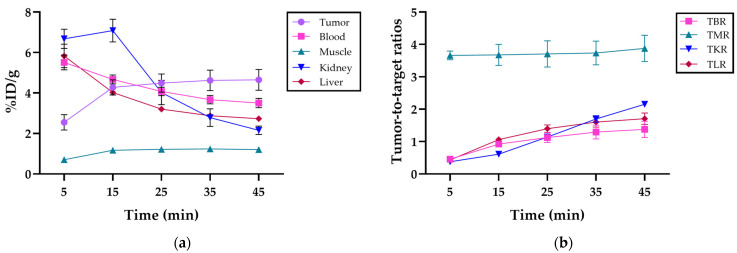
(**a**) Organ distribution in %ID/g of [^68^Ga]Ga-DOTAGA-Glu-(FAPi)_2_ in the tumor, blood, muscle, kidney, and liver, respectively, at 5, 15, 25, 35, and 45 min post-injection; (**b**) the tumor-to-blood ratio (TBR), tumor-to-muscle ratio (TMR), tumor-to-kidney ratio (TKR), and tumor-to-liver ratio (TLR) of [^68^Ga]Ga-DOTAGA-Glu-(FAPi)_2_ at 5, 15, 25, 35, and 45 min post-injection.

**Figure 5 diagnostics-14-02024-f005:**
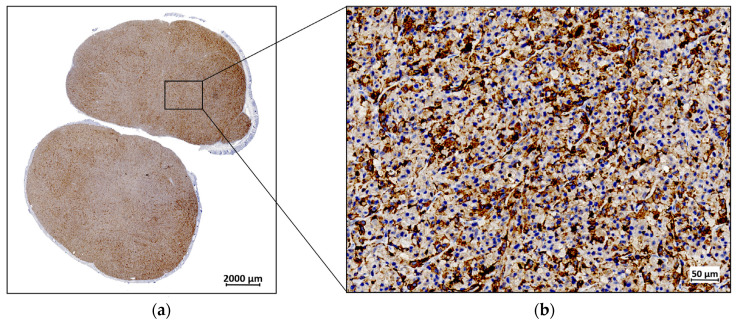
A representative example of fibroblast activation protein (FAP) expression by immunohistochemistry (IHC) in a U87MG resected tumor: (**a**) IHC staining for FAP at 1× magnification; (**b**) IHC staining for FAP at 40× magnification.

## Data Availability

Data are contained within the article or Appendix A.

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
