# Peer review of "[68Ga]Ga-DOTAGA-Glu(FAPi)2 Shows Enhanced Tumor Uptake and Theranostic Potential in Preclinical PET Imaging"

_diagnostics, 2024, doi:10.3390/diagnostics14182024_

Round 1

Reviewer 1 Report

Comments and Suggestions for Authors

It is an animal model paper that investigates the benefits of a new radiopharmaceutical for teragnosis. It is written in a clear and understandable way, the results are in accordance with the research and the conclusions. It is a suitable paper for publication.

Author Response

Overall response: Thank you for your much-appreciated comment and for taking the time to review our manuscript. 

Reviewer 2 Report

Comments and Suggestions for Authors

This is a very valuable research paper. In this study, [68Ga]-DOTAGA-Glu(FAPi)2 was synthesized and radiolabeled with 68Ga, and evaluated in vivo, which showed a potential tracer for diagnosis. However, the reviewer has the following concerns.

(1) For 68Ga labeling, it's very sensitive with pH, and what's this study final radiolabeling pH? For 68Ga-DOTA labeling, the pH is around 3.8, so I don't know why you use pH5.5 buffer.

(2) In Figure S2, why isn't there a UV-HPLC?

(3) The final formulated RCP is 88%, and why can't it be further purified to reach 90% or more for injection?

(4) For the PET imaging, what is the reason for the scan time being limited to 45 minutes?

(5) In figure 2, why is the TMR ratio relatively stable, which is inconsistent with the PET image display?

(6) Due to the absence of comparative studies with other tracers, definitive conclusions regarding the superiority of this tracer for treatment cannot be drawn. Furthermore, the study's scope is limited to 45min time point .

Comments on the Quality of English Language

The paper writing is professional and formal, and well-organized, which is easy to understand and the ideas are presented in a logical order, adhering to the rules of English grammar.

Author Response

Overall response: Thank you for taking the time to review our manuscript. We have provided detailed responses to your comments below, and the corresponding revisions or corrections have been highlighted in the resubmitted files.

Comment 1: For 68Ga labeling, it's very sensitive with pH, and what's this study final radiolabeling pH? For 68Ga-DOTA labeling, the pH is around 3.8, so I don't know why you use pH5.5 buffer.

Response 1: Thank you for your comment on this matter. 68Ga generator eluate used for labeling contains 0.1M HCl. The mixing ratio of 68GaCl3 eluate and ammonium acetate buffer (ca 6:1 v/v, respectively) is selected in a way that brings the pH of the final mixture into the optimal range for 68Ga labeling – between 3.5 and 4. In the particular production of [68Ga]Ga-DOTAGA-Glu-(FAPi)2 used for the imaging experiments, pH was not measured, but the mentioned mixing ratio has been used for years in our lab and is known to robustly produce pH between 3.5 and 4.

Comment 2: In Figure S2, why isn't there a UV-HPLC?

Response 2: Thank you for pointing this out. We updated the figure such that the UV traces are included for chromatograms in Figure S2, and the added Figure text has been marked in yellow.

Comment 3: The final formulated RCP is 88%, and why can't it be further purified to reach 90% or more for injection?

Response 3: Indeed, it should be possible to further optimize the labeling conditions of [68Ga]Ga-DOTAGA-Glu-(FAPi)2 to obtain RCP over 90% or over 95%. Given that all radioactive impurities were more hydrophilic than the target product, we reasoned that they would be eliminated from circulation with at least the same speed as the target product, and were unlikely to result in any off-target binding. Therefore, we decided not to optimize the synthesis further for this exploratory study.

Comment 4: For the PET imaging, what is the reason for the scan time being limited to 45 minutes?

Response 4: At the time of the experiments, the animal license under which this study was performed, permitted scanning of mice under anesthesia for a maximum of 1 hour. Given that the scan included both CT and PET components, we allocated 60 minutes for the scan, with 50 minutes for PET (displayed as the timing window of 45 min.) and 10 minutes for CT, which adhered to the constraints of the license (2021-15-0201-01041).

Comment 5: In figure 2, why is the TMR ratio relatively stable, which is inconsistent with the PET image display?

Response 5: Thank you for the comment. We assume that the reviewer refers to Figure 4.
The left quadriceps femoris was annotated on the CT scan as muscle tissue. While there may be a tendency towards higher tracer concentration at later time points in this tissue, it is offset by increased accumulation in the tumor tissue, thus yielding a relatively stable TMR.

Comment 6: Due to the absence of comparative studies with other tracers, definitive conclusions regarding the superiority of this tracer for treatment cannot be drawn. Furthermore, the study's scope is limited to 45min time point.

Response 6: Thank you for pointing this out. We agree that further investigation into the biodistribution and biokinetics of the therapeutic application (labeled with longer-lived isotopes) at later time points is essential to infer dosimetry and assess therapeutic potential accurately. However, this falls outside the scope of the current article, which suggests a theranostic companion to the 177-Lu labeled compound already undergoing clinical tests. In this current issue, focus is presenting this compound as a theranostic companion to the clinically tested ¹⁷⁷Lu-labeled version. Even though the scan duration is relatively short, a clear plateau is evident after 35 minutes, suggesting that peak tumor accumulation is reached at this timepoint.

Reviewer 3 Report

Comments and Suggestions for Authors

I appreciated your paper. Methodology is correct, results and discussion are presented in a clear manner.

Two minor remarks:

  • mean and standard deviation are well represented in figure, partially in the text but are not provided in the supplementary tables, where some statistical analysis comparing different times might be useful

  • the radiochemical nature of the peak 2 in HPLC is not explained, have the authors some ideas if this secondary compound can influence the radiopharmaceutical kinetics?

Author Response

Overall response: Thank you for taking the time to review our manuscript. We have provided detailed responses to your comments below, and the corresponding revisions or corrections have been highlighted in the resubmitted files.

Comment 1: mean and standard deviation are well represented in figure, partially in the text but are not provided in the supplementary tables, where some statistical analysis comparing different times might be useful

Response 1: We have added the mean and the SEM to the tables, which can be found in the supplementary Tables S1 and S2 marked with yellow.

Comment 2: the radiochemical nature of the peak 2 in HPLC is not explained, have the authors some ideas if this secondary compound can influence the radiopharmaceutical kinetics?

Response 2: We’d like to thank the reviewer for the observant question. DOTAGA-Glu-(FAPi)2 precursor contains three amide bonds between the glutamic acid linker, DOTAGA chelator, and UAMC-1110 moieties. Most probably, peak 2 represents a labeled impurity with one of the amide bonds hydrolyzed, i.e. either a molecule with a single UAMC-1110 moiety or an unconjugated DOTAGA (the latter is less likely judging by the small difference in retention time between peaks 2 and 3). Such an impurity was either present in DOTAGA-Glu-(FAPi)2 precursor used for labeling or formed from the precursor during the labeling reaction due to partial hydrolysis. Whether peak 2 is a FAPi monomer or simply a “free” DOTAGA, it will show less tumor binding than the full-sized tracer and faster elimination. Thus, provided that the relative abundance of such impurity is low enough (6% is deemed low enough), it should not distort the conclusions about the suitability of [68Ga]Ga-DOTAGA-Glu-(FAPi)2 for theranostic imaging of FAP-expressing tumors.